# Structural and Mechanical Properties of Ionic Di-block Copolymers via a Molecular Dynamics Approach

**DOI:** 10.3390/polym11101546

**Published:** 2019-09-23

**Authors:** Mengze Ma, Yao Fu

**Affiliations:** Department of Aerospace Engineering and Engineering Mechanics, University of Cincinnati, Cincinnati, OH 45221, USA; mamz@mail.uc.edu

**Keywords:** ionic block copolymer, molecular dynamics simulation, mechanical property

## Abstract

Polymerized ionic copolymers have recently evolved as a new class of materials to overcome the limited range of mechanical properties of ionic homopolymers. In this paper, we investigate the structural and mechanical properties of charged ionic homopolymers and di-block copolymers, while using coarse-grained molecular dynamics simulation. Tensile and compressive deformation are applied to the homopolymers and copolymers in the glassy state. The effect of charge ratio and loading direction on the stress-strain behavior are studied. It is found that the electrostatic interactions among charged pairs play major roles, as evidenced by increased Young’s modulus and yield strength with charge ratio. Increased charge ratio lead to enhanced stress contribution from both bonding and pairwise (Van der Waals + coulombic) interaction. The increase in the gyration of the radius is observed with increasing charge ratio in homopolymers, yet a reversed tendency is observed in copolymers. Introduced charge pairs leads to an increased randomness in the segmental orientation in copolymers.

## 1. Introduction

Polymerized ionic copolymers have recently evolved as a new class of materials to overcome the limited range of mechanical properties of ionic homopolymers, in order to achieve a combined high modulus and conductivity. Well defined nanostructures of long-range order from body-centered-cubic spheres to lamellae can be achieved through the self-assembly of incompatible polymers that are covalently attached to one another. Tunable morphology and domain size can be obtained by varying the chain length, interactions between block polymers, etc. For example, in the case of AB di-block copolymer, with reducing $f_{A}$, the fraction of A block, lamellar, cylindrical, gyroidal, and spherical ordered phases can gradually form from the initial disordered phase.

Block copolymers (BCP) in solid state can form robust films to accelerate the transport of ions and solvents within the self-assembled nanostructure channels, offering unique physiochemical advantages, such as the high electromechanical stability for specific ions within these channels. As the ionic BCPs combine the properties of ionic liquids and BCPs, there is a particular interest to apply them as electrolyte separator in solid-state lithium ion batteries. In the ionic BCPs, one block serves as the solid polymer electrolyte for the ion-conductive pathway, while the other block provides mechanical support [1,2]. Another important application is in the field of ionic electroactive polymer actuators. Upon the application of an electric field, the positive counterions move towards the negative electrode and negatively charged monomers fixed to the polymer backbones experience attractive force from the positive electrode. Previous studies found that this type of electroactive actuators demonstrate excellent electromechanical behavior, such as large displacement and fast response time, and thus receive great interest in the field of biomimetic technology [3,4,5,6,7,8,9].

In most of these applications, fast ion transportation [10,11] and high mechanical rigidity are the desirable properties [12]. Tyler et al. [13,14], Chakraborty [15], and Baeurle [16] have conducted very interesting studies by using the free energy of slightly deformed structures. Tyler found that the elastic moduli could be expressed as a function of interaction parameter χN, volume fraction f, and block asymmetry α. Chakraborty and Baeurle’s study involve random copolymers, yet again, the dependence of elastic constant on volume fraction has been identified. Besides, the disorder correlation along backbones has also been found to play a significant role [15]. Among recent computational studies on the mechanical properties of neutral BCPs, Zheng et al. studied the dynamic and static mechanical properties of tri-block copolymers (TPEs) with different morphology [17]. In this study, the deformation is applied at a temperature where the A-block is in glassy state and B-block in rubbery state. An increase in the mechanical behaviors has been found with increasing A-block volume and A-A block interaction. Zhang et al. discovered that pressure serves as the most crucial factor in determining the shear-under-pressure behaviors of hard-soft copolymer systems [18]. To quickly achieve the microphase segregation, soft-core potentials are often adopted in numerical simulations to first obtain ordered phase. The soft-core potentials are purely repulsive to capture the essential physics of self-assemble behavior [19,20]. Other techniques, such as lattice Monte Carlo (MC) [21], have also been attempted to achieve self-assembling that show faster simulation speed when compared to molecular dynamics (MD) simulations. 

The charged BCPs can possess different phase separation behavior from neutral BCPs. Sing [22], Pryamitsyn [23] and Goswami et al. [24] studied the influence of temperature and dielectric constant on the phase separation feature of ionic di-BCPs. The ion transport mechanism in ionic polymers also becomes a topic that attracts much attention. Sethuraman et al. has observed that five different ion transport mechanisms in lamellar BCPs and homopolymers (HPs), and that the mean square displacement of BCPs decreases with increasing charge ratio [25]. Ting et al. found that the ion transport of ionomer melt displays a different regime influenced by the strength of electric field and the ionic aggregate morphology [26]. However, there exist relatively few studies that attempt to understand the mechanical properties of the charged BCPs.

The purpose of this paper is to elucidate the mechanical responses of ionic HPs and di-BCPs with varying charge density. We employ a coarse-grained molecular dynamics (CGMD) approach to investigate the deformation behaviors of both ionic di-BCPs and HPs. The morphology at high temperature and glass transition behavior are first investigated. Uniaxial tensile and compressive loadings are then applied in the glassy state and mechanical responses up to post-yield regime are obtained. In Section 2, the computational model that involves a two-step equilibration strategy is introduced. The materials and simulation procedure are detailed in Section 3, followed by the results and discussion in Section 4.

## 2. Model and Simulation Details

All of the polymer chains of HPs are uniformly charged. In di-BCPs, only the A block is uniformly charged and the B block remains neutral. The degree of polymerization of a single polymer chain is fixed as 40 for both HPs and di-BCPs to avoid entanglement. A total of 1000 chains are generated. The charge ratio $f_{c,\mathrm{HP}}$ (the percentage of the polymer beads that are charged) of HPs is increased from 0% to 50%. The charge ratio of the A block polymers, $f_{c,A}$, in di-BCPs is also gradually increased from 0% to 50%. Since the fraction of A block, $f_{A}$, is 0.5, HPs and di-BCPs have the same number of charged ions when $f_{c,\mathrm{HP}}$ of HPs is half of $f_{c,A}$ of di-BCPs, e.g., 12.5% charged HPs and 25% charged di-BCPs possess equal number of charged ions. The charged beads and counterions have the charge value of 10 and -10, respectively. This value is chosen based on the computed reduced charge $q^{*}=\frac{q}{\sqrt{4\pi \sigma \varepsilon_{0}\epsilon }}.$ With $q=1.6e^{-19}\;C,\;\sigma =0.5\;nm$, and $\epsilon /k_{B}=\;298\;$, the estimated $q^{*}=10.6$. The number of counterions added to the system also varies in order to keep electrically neutral.

A two-step equilibration strategy [19,27,28] is employed to obtain the nanostructured BCPs. First, the polymers and counterions are randomly placed in the simulation box, with an initial number density set as $\rho^{*}$ = 0.8. An energy minimization step is performed to avoid monomer and counterions overlap. Subsequently, a soft-core potential, i.e. dissipative particle dynamics (DPD) type potential, is applied to quickly form nanostructured morphology. This type of soft-core DPD force field has been widely used to study colloidal particles and polymeric fluids [19,29]. This model allows for us to obtain ordered phase with much less computational time when compared to the Kremer-Grest (KG) model, which is usually employed to obtain refined structures. In the DPD model, the soft-core potentials are in the form, as follows:(1)$$U_{ij}\left(r\right)=\{\begin{array}{ll} \frac{a_{ij}{\left(r-r_{c}\right)}^{2}}{2r_{c}^{2}}, & r\leq r_{c}\\ 0, & r>\;r_{c} \end{array}$$
where $a_{ij}$ is the pairwise interaction parameter between polymer beads *i* and *j*. r represents distance of a pair of polymer beads and $r_{c}$ is the cut-off distance set as 1.68σ. The interaction strength between all components is set to be $a_{ij}=\;15.4\;k_{B}T$ except that the interaction between different monomer beads is set as $a_{AB}\;=\;50.0\;k_{B}T$. NPT ensemble is employed where the Noose-Hoover thermostat and Berendsen barostat are used to maintain a constant pressure of $P^{*}\;=\;4.5$ and temperature of $T^{*}=1.0$. The temperature friction coefficient is set as 0.5. The equilibration time is set as 5e5 time steps with a time step size of $dt^{*}\;=\;0.02$. The choice of soft-core model parameters is to ensure the match between the soft-core and hard-core (KG) model in predicting the densities and other structural characteristics. Interested readers are referred to [19], where detailed discussion on choosing the model parameters can be found.

After the ordered phase di-BCP is obtained, the Kremer-Grest (KG) bead spring model with hard-core potential is applied for a second step equilibration in order to obtain refined structure. The KG model includes a shifted Lennard-Jones (LJ) potential that is used to describe the non-bonded interaction, as follows,
(2)$$U_{ij}\left(r\right)\{\begin{array}{cc} 4\epsilon \left[{\left(\frac{\sigma }{r}\right)}^{12}-{\left(\frac{\sigma }{r}\right)}^{6}\right]S\left(r\right),\; & \;r\leq r_{c}\\ 0\;, & r>r_{c}\; \end{array}$$
where σ is the unit length, ϵ is the unit energy of this model, and S(r) is the shifting function to ramp the energy smoothly to zero at the cut-off distance [30]. The interaction strength between all of the components is set as ϵ = 1.0 *kT*, except that the strength between A-B pair of beads is set as ϵ = 0.1 *kT*, and $r_{c}=\;2.5\;\sigma$.

Adjacent polymer beads on a chain are bonded via a finite extensible nonlinear elastic (FENE) potential, as follows,
(3)$$U_{\mathrm{FENE}}=-0.5KR_{0}^{2}\ln\left[1-{\left(\frac{r}{R_{0}}\right)}^{2}\right]+4\epsilon \left[{\left(\frac{\sigma }{r}\right)}^{12}-{\left(\frac{\sigma }{r}\right)}^{6}\right]+\epsilon$$
where K = 30$\epsilon \sigma^{-2}$ and the maximum possible length of the bond is $R_{0}=1.5\sigma$.

The pairwise coulombic interaction is also turned on in the hard-potential model to incorporate the electrostatic interaction between charged polymer beads and counterions into the total potential. Coulombic interaction can be expressed, as follows:(4)$$U_{Coul}\left(r\right)=\{\begin{array}{ll} \frac{q_{i}q_{j}}{4\pi \varepsilon_{0}\varepsilon_{r}r}, & r\leq r_{c}\\ 0, & r>r_{c} \end{array}$$
where $\varepsilon_{0}$ is the permittivity of free space, $\varepsilon_{r}$ is relative permittivity (dielectric constant), and $q_{i},q_{j}$ are charge value of components i and j. Dielectric constant, $\varepsilon_{r}$, is set to be 10 to simulate the effects of moderately polarized polymeric materials. The cut-off distance for LJ interaction is set as 2.5. The long-range columbic interaction is solved while using the PPPM technique. Since we are not restricted to a particular type of polymers, reduced unit will be used throughout the paper, so that $r^{*}=r/\sigma$, $T^{*}=k_{B}T/\epsilon$, and $t^{*}=t/\tau$, where $\tau ={\left(m\sigma^{2}/\epsilon \right)}^{1/2}.$

The two-step strategy is also applied to generate the initial structure of HPs with the interaction strength a and ϵ between all of the components set to be same. After the equilibrated structure has been obtained using the KG model, HPs and BCPs are both quenched down to $T^{*}\;=\;0.3$ at a cooling rate of $\Delta T^{*}/\Delta t^{*}=1e^{-7}$. At $T^{*}\;=\;0.3$, which is below the glass transformation temperature, the deformation tests are conducted. Uniaxial tensile and compressive loadings are conducted by changing the box length under a constant engineering strain rate of ${\dot{\varepsilon }}^{*}$ = 0.002, whereas the lateral pressures are kept as zero. The stress-strain behaviors of di-BCPs and HPs of varying charge ratio are examined up to a strain of ε=0.1. Tension and compression simulations are both performed along two coordinate directions for BCPs to account for the effect of anisotropy. It is worth mentioning that the KG model is used for all of the subsequent CGMD simulations for understanding the structural and mechanical properties. At each condition, a minimum of three tests have been conducted to ensure repeatability.

## 3. Results

### 3.1. Morphologies and Number Distribution

The simulation boxes of HPs and di-BCPs equilibrated at $T^{*}=\;1.0$, under the anisotropic NPT ensembles are demonstrated in Figure 1 and Figure 2 (top panel), respectively. At first, the pressures in the x, y, and z directions are adjusted in a coupled manner. The anisotropic nature of the system leads to a non-zero averaged pressure in three directions. Therefore, a further equilibration under NPT ensemble is carried out where the pressures in the *x*, *y*, and *z* directions are separately adjusted. The cubic simulation boxes of di-BCPs change their shape due to the anisotropic nature of the systems. The lamellar phase increases in length in the perpendicular to the interface direction during equilibration, accompanied by the shrinkage along the lateral side, similarly to what has been observed by Ryu et al [27]. The ratio of box size in the three directions is approximately 5:3:3 at $T^{*}\;=1.0$ (Figure 2), where the elongated direction is the direction perpendicular to the lamella layer. In contrast, the simulation box of HPs remains approximately cubic during the equilibration (Figure 1). The ratio of box size in the three directions does not significantly change with increasing charge ratio. The counterions locate mainly in the A block adjacent to charged monomers, as seen in Figure 2 (bottom panel). The domain width of A block increases gradually with charge ratio due to the increasing number of counterions that reside in A block.

It is worth mentioning that the system is simulated while using the hard-core potential for sufficiently long time until no significant change in morphology or instability is observed. During the later stage of equilibration, the radius of gyration $R_{g}$ stabilizes with small fluctuation and no tendency of $R_{g}$ increase or decrease with time is observed. The structural and mechanical properties are analyzed after the equilibration and quench process. However, given the limitation in the time scale of MD simulation, it is still possible that the system reaches a local equilibrium state instead of a full equilibrium state. The equilibrium morphology of the charged BCPs has been an active research topic in recent years. It has been theoretically predicted that the morphology diagram is highly asymmetric in charged copolymers. The counter-ions tend to be located in the domains containing the charged ions [31]. Sing and Pryamitsyn et al. discussed that nanostructures can be tuned through the highly asymmetric charge cohesion effect [22,23]. It has been demonstrated that different phases from bcc spheres to hcp cylinders and lamellae can be formed in a charged BCP of the A block fraction $f_{A}=0.25$. This is drastically different from the prediction of phase diagram of neutral BCPs. The change in the phase diagram of the charged polymers from that of the uncharged one strongly depends on the strength of the electrostatic cohesion that is indicated by the parameter $\Gamma =e^{2}/\left(8\pi \varepsilon_{0}\varepsilon_{r}kT\sigma \right)$. The parameter Γ expresses the magnitude of the Coulombic potential between two charges of strength *e* at their closest distance, in units of thermal energy $k_{B}T$. The phase diagram has not been found to alter much at a low value of Γ.

The parameter Γ is estimated approximately as 5 when $\epsilon_{r}$ equals 10. In this case, the phase diagram is shifted vertically instead of leftwards, as predicted by Sing et al. [22]. Therefore, only the segregation strength dependence is varied, but volume fraction dependence is kept in the similar range as that in the neutral copolymers. This possibly explains that the lamellar microstructure remains stable, even with increasing charge density at $f_{A}=0.5$ in our study.

### 3.2. Radial Dstribution Function

Radial distribution function g(r) is the function used to describe the molecular-level structure of the system. The g(r) for pairs of beads is given, as follows,
(5)$$g\left(r\right)=\;\frac{\sum_{j=1}^{N_{c}}N_{j}\left(r\pm \frac{\Delta r}{2}\right)}{\Omega \left(r\pm \frac{\Delta r}{2}\right)\rho N_{c}}$$
where $N_{j}\left(r\pm \frac{\Delta r}{2}\right)$ is the number of charged/neutral beads residing in the spherical shell of radius r of bead *j*, $\Omega \left(r\pm \frac{\Delta r}{2}\right)$ is the volume of the spherical shell of thickness Δr, ρ is the averaged density of charged/neutral beads, and $N_{c}$ is the total number of bead *j*. In Figure 3, the radial distribution function of neutral monomer-neutral monomer, charged monomer-charged monomer, charged monomer-counterion, and counterion-counterion pair are computed for both HPs and di-BCPs with varying charge ratio, denoted as $g_{nm-nm}$, $g_{cm-cm}$, $g_{cm-ci}$, and $g_{ci-ci}$, respectively. Only the neutral monomers A block in the di-BCPs are counted in the calculation. In $g_{nm-nm}$, the primary peak locates at around $r^{*}=1.0$. Its intensity decreases with charge ratio most apparently at $f_{c}=50\%$ and it shifts slightly to the higher radius. Instead, the peak locating at $r^{*}=1.0$ for charged pairs ($g_{cm-cm}$, $g_{ci-ci}$) become stronger. Interestingly, the primary peak of $g_{cm-ci}$ decreases with charge ratio, possibly due to the increased electrostatic screening effect. The charge ratio dependent radial distribution functions g(r) are very similar between HPs and BCPs in terms of the peak location and relative strength. However, g(r) does not converge to unity with increasing r in BCPs as in HPs mainly due to the microphase separation and the charged atoms located only in the A block.

### 3.3. Glass Transition Temperature

Figure 4 demonstrates density versus temperature relation of both HPs and di-BCPs with varying charge ratio. Glass transition temperature ($T_{g}$) can be obtained by the change in the thermal expansion coefficient [32,33,34]. $T_{g}$ of both charged HPs and BCPs gradually increase with charge ratio. These results are in agreement with our previous study in the ionic homopolymer system, where an increase in the glass transition temperature of charged HPs was found as compared to their neutral counterpart [32,33]. At the same charge ratio, the $T_{g}$ of BCPs is lower than that of HPs due to the lower $T_{g}$ of B block. As the number of charged ions in HPs is same as that in BCPs when $f_{c,A}$ is half of $f_{c,HP}$, the $T_{g}$ of 12.5% charged HPs is close to that of 25% charged BCPs, and the $T_{g}$ of 25% charged HPs is close to that of 50% charged BCPs. Both blocks of the lamellar phase are in their glassy state at $T^{*}=0.3$.

### 3.4. Mechanical Properties of Copolymers 

Figure 5 demonstrates the stress-strain behaviors of ionic HPs and BCPs under tensile and compressive deformation at $T^{*}=0.3$. For the di-BCPs, perpendicular and parallel directions will be used in the following context to refer to the directions perpendicular and parallel to the interfaces. The BCPs deformed along the two different directions do not demonstrate major change in behaviors, and therefore only those in parallel direction are shown in Figure 5. The stress σ is computed via virial expression.
(6)$$\sigma_{IJ}=-\left\{\frac{\sum_{k}^{N}m_{k}v_{kI}v_{kJ}}{V}+\frac{\sum_{k}^{N’}r_{kI}f_{kJ}}{V}\right\}$$
where *N* is the number of atoms in the simulation box and *N’* includes the number of ghost atoms outside the central box. *V* is the system volume, $r_{kI}$ and $f_{kJ}$ are the *I* and *J* component of position r and force **f** of atom *k*, respectively.

The linear elastic regime appears within a small strain range, and then the system enters the nonlinear elastic and plastic regime. A strong dependence on the charge ratio can be observed for both HPs and BCPs. Introducing charged beads also slightly extends the elastic deformation regime.

Elastic modulus is obtained by the linear fitting of elastic region (ε <0.025 ), and yield strength is measured by the 0.2% offset method. Figure 6 plots the extracted values. Under both tension and compression along the two directions, increased mechanical properties (Young’s modulus and yield strength) are found with increasing charge ratio in the HPs and BCPs, most obviously from neutral to 12.5% charged in HPs and from neutral to 25% charged in BCPs. In the ionic HPs, the improvement in mechanical properties is not prominent when the charge ratio is further increased from 25% to 50%. As expected, the Young’s modulus and yield strength under compression is higher than that under tension.

Among all of the materials systems studied, the lamellar BCPs with the highest charge ratio ($f_{c,A}=50\%$) demonstrates the best mechanical properties, possessing the highest yield stress among all charge ratios and morphologies. The yield strength and Young’s modulus reach $\sigma_{y}^{*}=\;$1.08 and $E^{*}=36.46$ under tension, and $\sigma_{y}^{*}=$1.14 and $E^{*}=\;$39.59 under compression when deformed in the parallel direction. This leads to the yield strength on the order of 100 MPa and Young’s modulus in the order of 1 GPa, assuming that σ=0.5 nm and $\epsilon /k_{B}=\;298\;K$.

It is worth mentioning that even though no obvious difference is observed when the ionic BCPs are deformed along the perpendicular and parallel directions up to the strain of 0.1, a larger tensile deformation (ε=0.3) demonstrates that the loading direction does play a role in the deformation of BCPs (Figure 7). When stretched in the perpendicular direction, there is a clear strain hardening, which is hard to observe when stretched in the parallel direction. The anisotropic mechanical properties are very likely related to the chain and bond alignment, as will be demonstrated in the following sections.

### 3.5. Radius of Gyration during Deformation

To understand the conformation of the polymer molecules of different morphologies during deformation, mean square radius of gyration, $\langle R_{g}^{2}\rangle$ is calculated, as follows,
(7)$$\langle R_{g}^{2}\rangle \;=\;\langle \frac{1}{N_{p}}\overset{N_{p}}{\underset{i=1}{\sum }}{\left(r_{i}-r_{G}\right)}^{2}\rangle$$
where $N_{p}$ is the degree of polymerization of the molecule, $r_{i}=\left(x_{i},\;y_{i},\;z_{i}\right)$ is the position of polymer beads, and $r_{G}=\left(x_{G},\;y_{G},\;z_{G}\right)$ is the center of mass, and 〈⋯〉 denoting a statistical average over all of the chains. Its perpendicular and parallel components $\langle R_{g∥}^{2}\rangle$ and $\langle R_{g⊥}^{2}\rangle$ are given as follows if the *x* direction is the parallel direction and the perpendicular direction lies on the *y-z* plane:(8)$$\langle R_{g∥}^{2}\rangle \;=\;\langle \frac{1}{N_{p}}\overset{N_{p}}{\underset{i=1}{\sum }}{\left(x_{i}-x_{G}\right)}^{2}\rangle$$(9)$$\langle R_{g⊥}^{2}\rangle \;=\;\langle \frac{1}{2N_{p}}\overset{N}{\underset{i=1}{\sum }}{\left(y_{i}-y_{G}\right)}^{2}+{\left(z_{i}-z_{G}\right)}^{2}\rangle$$

The gyration of radius $\langle R_{g}\rangle$, as well as its components $\langle R_{g∥}\rangle$ and $\langle R_{g⊥}\rangle$ are defined as $\langle R_{g}\rangle =\sqrt{\langle R_{g}^{2}\rangle }$, $\langle R_{g∥}\rangle =\sqrt{\langle R_{g∥}^{2}\rangle }$, and $\langle R_{g⊥}\rangle =\sqrt{\langle R_{g⊥}^{2}\rangle }$.

Here, the calculated $\langle R_{g}^{*}\rangle$ of neutral HPs has the value of ~3.0, which is a little larger than the estimation from flexible ideal chains given by $\sqrt{b^{2}N/6}\;\left(\approx 2.57\right)$, where b is the monomer length and N the degree of polymerization. $\langle R_{g}^{*}\rangle$ of ionic HPs gradually increase with charge ratio (Figure 8). Under tensile and compressive deformation, $\langle R_{g}^{*}\rangle$ only slightly varies, when its components $\langle R_{g∥}^{*}\rangle$ and $\langle R_{g⊥}^{*}\rangle$ show a near linear relation with applied strain. Ionic BCPs have significantly increased values of $\langle R_{g}^{*}\rangle$ as compared with HPs. Its parallel and perpendicular components are no longer comparable but show a major difference in the direction perpendicular and parallel to the interface (Figure 9), which indicates the chains aligning perpendicular to the interface. Interestingly, the introduced charged beads reduce $\langle R_{g}^{*}\rangle$, in contrast to ionic HPs.

When loaded along the perpendicular direction of lamellar interface, the chains tend to be stretched, thus having larger $\langle R_{g}^{*}\rangle$. A reduced $\langle R_{g}^{*}\rangle$ is expected when compressed along the same direction (Figure 9). On the contrary when the lamellar domains are stretched and compressed parallel to the interface, a reduced and increased $\langle R_{g}^{*}\rangle$ is expected, respectively, as observed from Figure 10. In both cases of BCPs, the components $\langle R_{g∥}^{*}\rangle$ and $\langle R_{g⊥}^{*}\rangle$ show a near linear relation with applied strain, similar as that observed in HPs.

### 3.6. Second Order Legendre Polynomials

Bonds that connect the monomer beads can play a major role in the mechanical performance of the polymeric material, second order Legendre polynomials $\langle P_{2}(\cos\theta )\rangle$ is thus calculated, as follows,
(10)$$\langle P_{2}(\cos\theta )\rangle \;=\;\left(3\langle \cos^{2}\theta \rangle -1\right)/2$$
where θ is the angle between the direction of two adjacent monomer beads in the chain and the reference direction, chosen as the deformation direction. The value of $\langle P_{2}\left(\cos\theta \rangle \right)$ ranges from −0.5 to 1. When $\langle P_{2}(\cos\theta )\rangle \;=\;$1 and −0.5, it means that the average bond orientation in the system is parallel and perpendicular to the tensile or compressive direction, respectively. As shown in Figure 11, ionic HPs have random distributed bonds initially, which under tension and compression preferentially align along and perpendicular to the loading direction, respectively. In BCPs, the bonds tend to align toward the direction perpendicular to lamellar interface initially, as reflected by a nonzero $\langle P_{2}(\cos\theta )\rangle$, which has a positive value when deformed perpendicular to the interface and negative when deformed in parallel (Figure 12). $\langle P_{2}(\cos\theta )\rangle$ increases under tension and decreases under compression in all phases, which shows that the bonds gradually align to the direction parallel to the tensile direction or perpendicular to the compressive direction, respectively. The absolute change in $\langle P_{2}(\cos\theta )\rangle$ value with deformation has been found to be very close between the parallel and perpendicular directions. Yet, the perpendicular loading direction can more easily reorient the chain segments, as observed from the slightly more significant change in the value of $\langle P_{2}(\cos\theta )\rangle$ with loading. The change in $f_{c}$ has been observed to shift the values of $\langle P_{2}(\cos\theta )\rangle$ closer to zero (Figure 12), which indicates more randomly ordered chain segments associated with the increasing number of counterions, in agreement with the analysis from radius of gyration.

## 4. Discussion

Introduced charged beads and counterions have been demonstrated to affect the configurational and mechanical properties of HPs and BCPs. With an increase in charge ratio, the radius of gyration of HPs gradually increase, same as the prediction from previous theoretical study on a single charged polymer chain that shows a stiffened chain due to the repulsive intrachain interaction [35]. In the charged polymer–counterion melt/glass, the types of interaction become more complicated in that the counterions also interact with polymer monomers via Van der Waals force described by the LJ potential. Thus, it is difficult to attribute the swollen polymers chains purely to the electrostatic effect. The charge of both monomers and counterions in the 12.5% charged HPs is set as zero and the system is equilibrated for another 1e7 timesteps until the radius of gyration fluctuates at a constant value to clarify the influence of Van der Waals and electrostatic effects. It was found the $\langle R_{g}^{*}\rangle$ increases from around 3.1 to 3.2 when the charge is set to zero, i.e., when only the Van der Waals interaction takes place. This is most likely due to the increased intrachain and interchain attraction through crosslinked charged monomers via counterions, which reduces the gyration of radius. The variation of $\langle R_{g}\rangle$ is observed to decrease in BCPs with charge ratio, still most likely attributed to the increased intrachain and interchain attraction, which prevents the chains from aligning perpendicular to the interface direction, as evidenced by the increased $\langle R_{g⊥}\rangle$ with charge ratio (Figure 9).

The variation of $\langle R_{g}\rangle {}_{∥}$ and $\langle R_{g}\rangle {}_{⊥}$ and $P_{2}\left(\cos\theta \right)$ demonstrate a linear dependence with applied strain (Figure 8, Figure 9 and Figure 10), showing similar dependence at a different charge ratio. To further understand the strengthening origin with increasing charge ratio, energy decomposition into bonded, Van der Waals, and coulombic contribution are conducted and demonstrated in Figure 13. Two neutral systems, 0% charged HP and 0% charged BCP, are compared with two charged systems, 12.5% HP and 25% BCP, during tensile loading. The two charged systems contain the same number of charged ions to enable a direct comparison. The results show that the neutral and charged systems have close values in bonded and Van der Waals energy, respectively. Interestingly, the introduction of charged beads lowers the bonded energy. Stress decomposition reveals that the bonded contribution in all cases is compressive stress, which indicates that bonds are in compression. Therefore, in charged system, the bonds in compressive state are getting closer to the equilibrium length (Figure 13). Van der Waals energy is also lower in charged systems as compared with that in neutral systems. Coulombic interaction has a lower absolute value than Van der Waals considering only 25% monomers are charged. The Van der Waals energy has a relatively sharper increase at lower strain than at larger strain, which is consistent with the tensile loading of polymeric materials at similar strain range [36], despite some difference in the physical models. Different from Van der Waals energy, the coulombic energy has a close-to-linear relation with applied strain. The total pair energy, i.e., the sum of Van der Waals and coulombic energy, demonstrates the energy that was stored in charged systems is higher than that in neutral systems.

The stress decomposition into bonded and non-bonded interactions is further demonstrated in Figure 14. In Equation (6), $f_{k}$ can be decomposed into bonded and non-bonded (pair) terms, i.e., $f_{k}=f_{k,bond}+f_{k,pair}$, and the corresponding stress contributions are thus obtained. The first term related to kinetic energy is not included in the calculated stress that is shown in Figure 14, and hence only the second term in the curly bracket is considered,

Even though the bonded energy plot shows no clear tendency, the stress from bonded contribution has a clear initial increase, followed by slight decrease or flattening. It is worth mentioning that the plots are made relative to the values at zero deformation, and the absolute bonded stress remain at negative values through the loading process. The bonded stress has greater contribution in charged systems than in neutral systems, similar to the case of pair stress. Interestingly, the enhancement of stress from bonded and non-bonded contribution are comparable with increasing charge ratio, and lead to the increased stress values, as demonstrated in the $\sigma_{\mathrm{bond}}+\sigma_{\mathrm{pair}}$ plot. The increase in bonded and pair stress have also been observed in the 25% charged HP and 50% charged BCP (figures not shown). In addition, it is found that the HPs and BCPs that possess the same number of ions have similar enhancement in mechanical properties (Figure 14).

The final remark is that our study found the anisotropy of charged BCP is not apparent at small deformation. The preliminary study shows that the effect might become apparent at larger deformation (Figure 7). Therefore, more detailed study of the anisotropy effect will be considered for future work. Of equivalent interest are the viscoelastic behaviors at rubbery state and the effect of different loading mode, e.g., shearing, which will also be considered in our work in the near future.

## 5. Conclusions

In this study, CGMD simulations are performed to investigate the structural and mechanical behaviors of ionic HPs and di-BCPs while using CGMD simulation. The materials are evaluated in terms of structural, configurational, and mechanical properties, which have demonstrated quite distinct features as a result of the varying charge ratio and loading directions. Some main findings are summarized, as below:A strong dependence of mechanical properties on increasing charge ratio have been observed in ionic HPs up to 25% and in BCPs up to 50% charged A block. Further increasing the charge ratio from 25% to 50% in HPs does not result in the improvement of mechanical properties.Anisotropy in mechanical properties in the lamellar BCPs is not apparent when deformed to a strain level of 0.1 along the directions parallel and perpendicular to the microphase interface. When stretched up to a strain level of 30%, the mechanical responses start to show large discrepancies, which is expected as the weak interface between A and B block orients differently with respect to the two loading directions.Chain segments preferentially align perpendicular to the microphase interface in ionic di-BCPs, in contrast to the random distribution in ionic HPs. The radius of gyration $\langle R_{g}\rangle$ increases with increasing charge ratio in HPs. However, $\langle R_{g}\rangle$ shows a reverse tendency with the charge ratio in BCPs. A linear dependence of the parallel and perpendicular components of the radius of gyration and bond orientation on the applied strain is observed under all loading conditions of HPs and BCPs.HPs do not show a significant change in the radius of gyration, whereas di-BCPs have an increase in $\langle R_{g}\rangle$ when stretched perpendicular to the interface or compressed parallel to the interface. BCPs have a decrease in $\langle R_{g}\rangle$ when stretched parallel to the interface or compressed perpendicular to the interface.

## Figures and Tables

**Figure 1 polymers-11-01546-f001:**
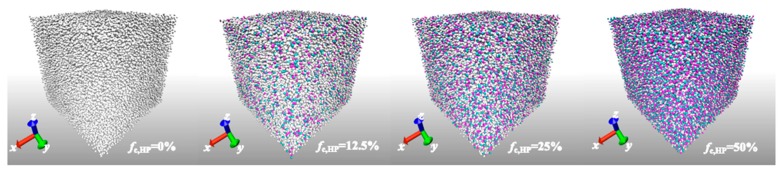
Morphology of ionic homopolymers (HPs) with varying charge ratio $f_{c,\mathrm{HP}}$ from 0% to 50%, where the neutral monomers are in white, the charged monomers in red and charged counterions in cyan.

**Figure 2 polymers-11-01546-f002:**
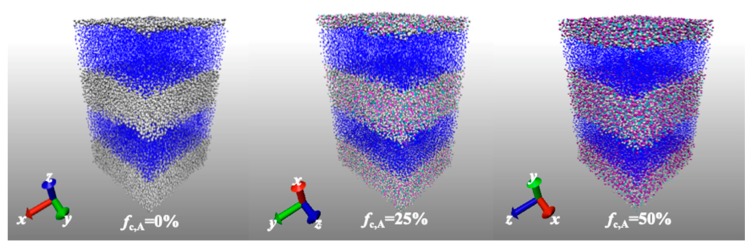
(**Top panel**) Morphology of lamellar ionic Block copolymers (BCPs) with varying charge ratio of A block, $f_{c,A}$ from 0% to 50%, where the neutral A block is in white and B block in blue, the charged A block in red and charged counterion in cyan; and (**bottom panel**) number distribution of A block monomer and charged counterions along the height of the simulation box of the corresponding systems equilibrated at *T^*^* = 1.0.

**Figure 3 polymers-11-01546-f003:**
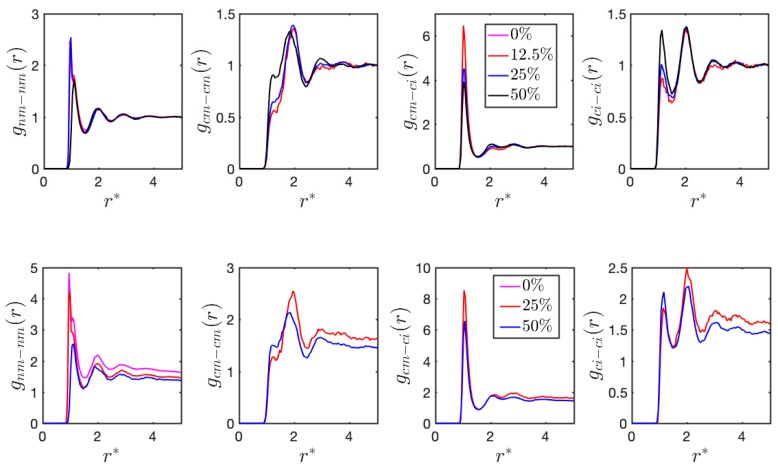
Radial distribution function of neutral monomer-neutral monomer ($g_{nm-nm}$), charged monomer-charged monome $g_{cm-cm}$, charged monomer-counterion $g_{cm-ci}$, and counterion -counterion $g_{ci-ci}$ pair of (top panel) HPs and (bottom panel) di-BCPs at *T^*^* = 1.0.

**Figure 4 polymers-11-01546-f004:**
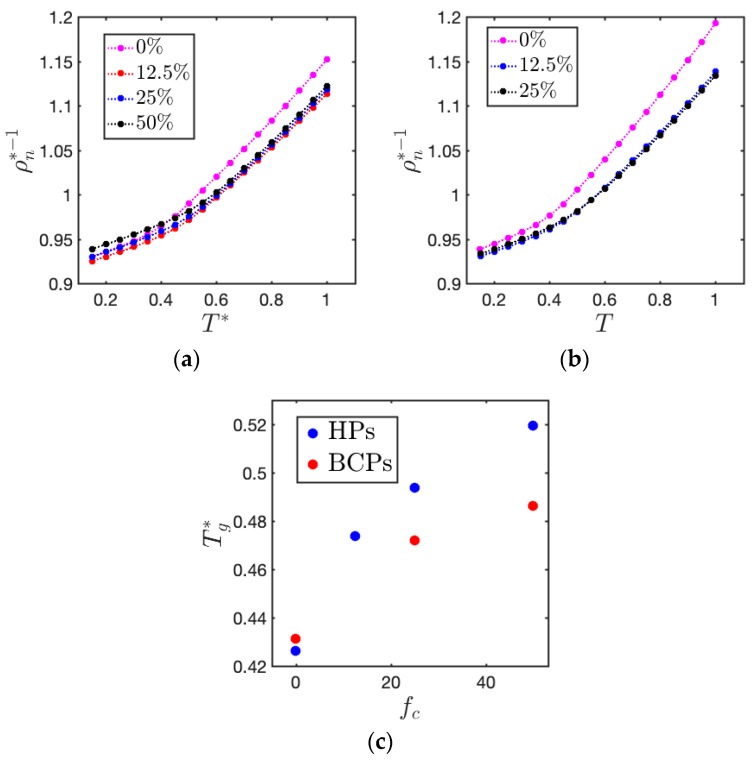
Specific volume of the system ($\rho_{n}^{*-1}$) as a function of temperature of (**a**) HPs and (**b**) BCPs at different charge ratio *f_c_*, and (**c**) extracted $T_{g}^{*}$ as a function of *f_c_*.

**Figure 5 polymers-11-01546-f005:**
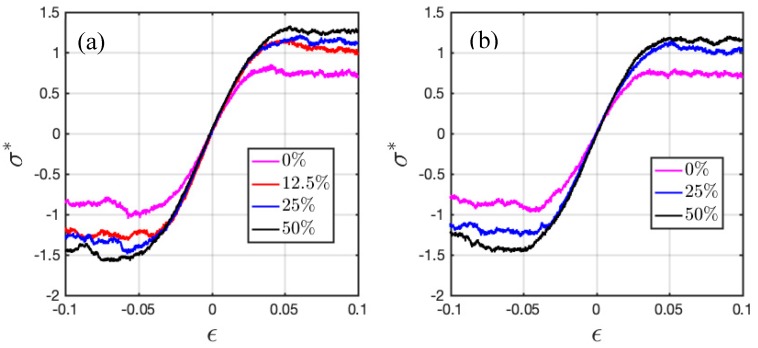
Stress-strain behavior of (**a**) ionic HPs and (**b**) di-BCPs deformed in the parallel to the interface direction at $T^{*}$ = 0.3.

**Figure 6 polymers-11-01546-f006:**
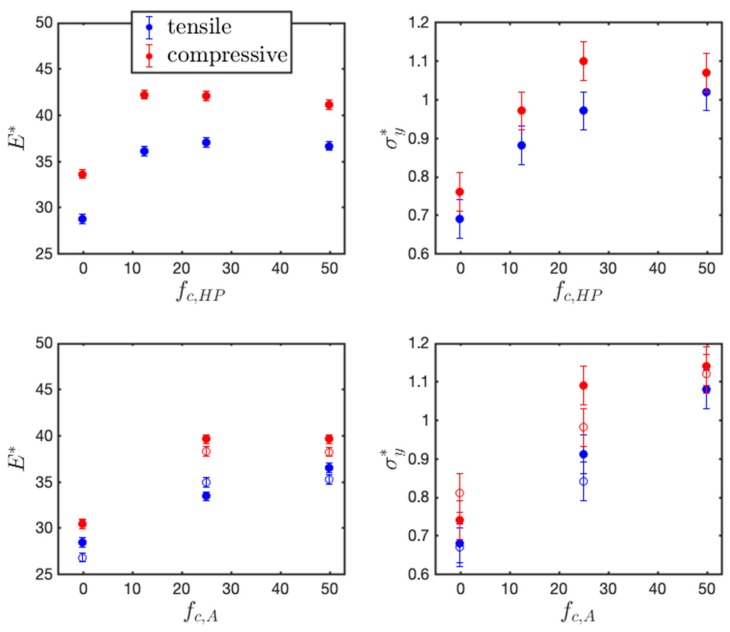
Extracted elastic modulus (E and yielding stress ($\sigma_{y}$) of (**top panel**) HPs and (**bottom panel**) BCPs at different charge ratio at $T^{*}$ =0.3 (filled symbol in parallel direction and open symbol in perpendicular direction).

**Figure 7 polymers-11-01546-f007:**
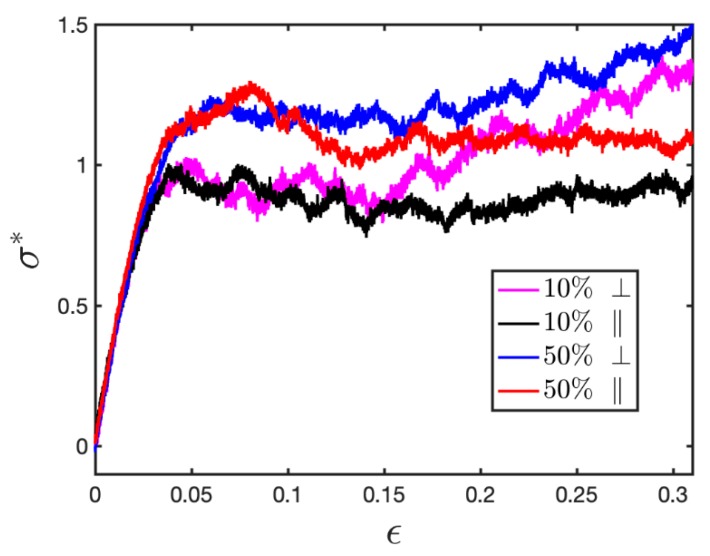
Stress-strain relation of lamellar ionic di-BCPs stretched perpendicular and parallel to the interface.

**Figure 8 polymers-11-01546-f008:**
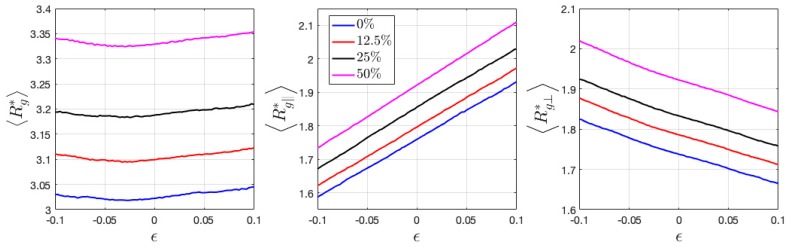
Variation of radius of gyration $\langle R_{g}^{*}\rangle$, and its parallel and perpendicular components $\langle R_{g∥}^{*}\rangle$ and $\langle R_{g⊥}^{*}\rangle$ during the deformation of ionic HPs. The parallel direction is chosen as the deformation direction.

**Figure 9 polymers-11-01546-f009:**
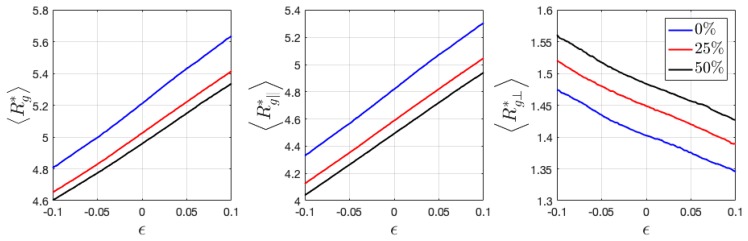
Variation of radius of gyration $\langle R_{g}^{*}\rangle$, and its parallel and perpendicular components $\langle R_{g∥}^{*}\rangle$ and $\langle R_{g⊥}^{*}\rangle$ when deformed in the perpendicular to interface direction in the ionic BCPs. The parallel direction is chosen as the deformation direction.

**Figure 10 polymers-11-01546-f010:**
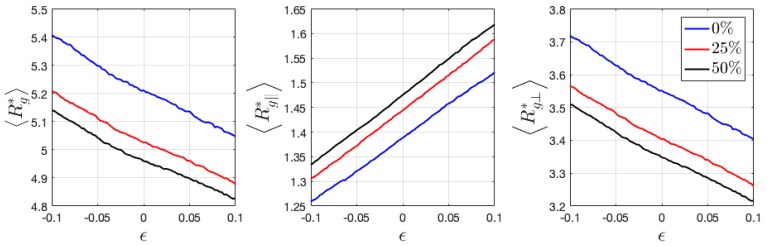
Variation of radius of gyration $\langle R_{g}^{*}\rangle$ and its parallel and perpendicular components $\langle R_{g∥}^{*}\rangle$ and $\langle R_{g⊥}^{*}\rangle$ when deformed in the parallel to interface direction. The parallel direction is chosen as the deformation direction.

**Figure 11 polymers-11-01546-f011:**
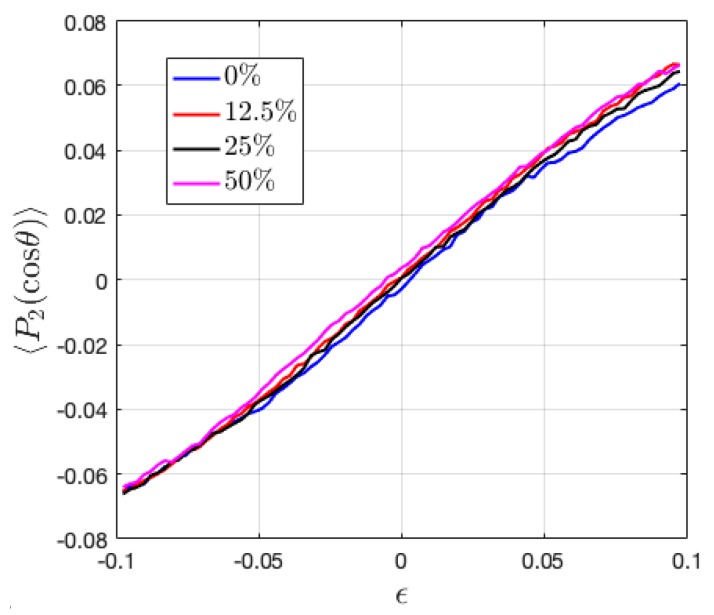
The change of bond orientation during the tension and compression of ionic HPs with varying charge ratio $f_{c,HP}$.

**Figure 12 polymers-11-01546-f012:**
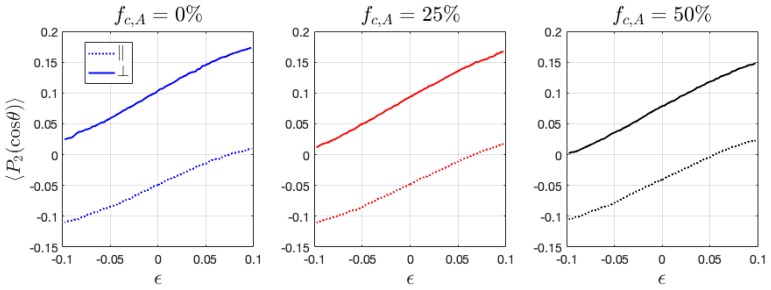
The change of bond orientation during the tension and compression of ionic BCPs with varying charge ratio $f_{c,A}$.

**Figure 13 polymers-11-01546-f013:**
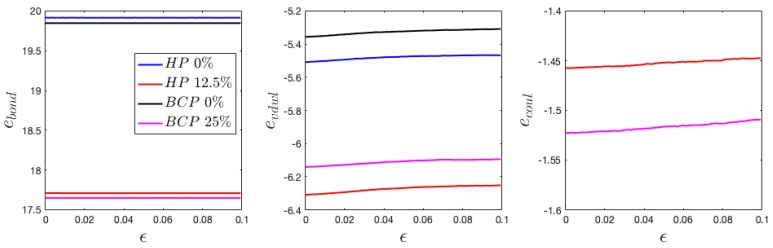
The change of bonded (*e*_bond_), Van der Waals (*e*_vdwl_), and coulombic energy (*e*_coul_) during tensile loading perpendicular to the interface.

**Figure 14 polymers-11-01546-f014:**
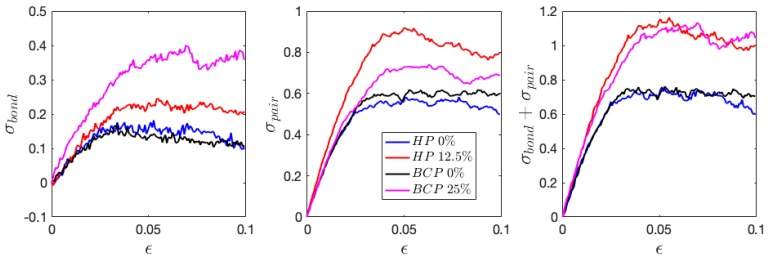
The change of bonded and pair (Van der Waals + coulombic) stress contribution relative to the initial values at zero deformation during tensile loading perpendicular to the interface.

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
