# Peer review of "Structural and Mechanical Properties of Ionic Di-block Copolymers via a Molecular Dynamics Approach"

_polymers, 2019, doi:10.3390/polym11101546_

Round 1
Reviewer 1 Report
The manuscript titled “Structural and Mechanical Properties of Ionic Di-block Copolymers via a Molecular Dynamics Approach” used coarse grained molecular dynamics simulations to investigate the structural and mechanical behaviors of ionic homopolymers and di-Block copolymers. Overall, the manuscript is well designed and executed. Please review the manuscript for grammatical errors.
Reviewer 2 Report
Comments and questions:
The section 2 and 3 and should be rewritten. It should be well orginazed. How is the temperature rate determined in Figure 4. How is the stress calculated? No overshoot is seen in Figure 5. why?Author Response
Please see the attachment.

Round 2
Reviewer 2 Report
1. please add a formula or reference of the reduced charge calculation.
2. why is the cut-off set as 1.68?
3. Is the soft-core potential applied on each pair of CG beads or the CG beads on
different chains?
4. Why the pressure is 4.5?
5. please check formula 2.Is it the same as the original KG model?
Round 3
Reviewer 2 Report
The authors have addressed all the questions.